# Lactate Promotes Hypoxic Granulosa Cells’ Autophagy by Activating the HIF-1α/BNIP3/Beclin-1 Signaling Axis

**DOI:** 10.3390/genes16010014

**Published:** 2024-12-26

**Authors:** Yitong Pan, Gang Wu, Min Chen, Xiumei Lu, Ming Shen, Hongmin Li, Honglin Liu

**Affiliations:** College of Animal Science and Technology, Nanjing Agricultural University, Nanjing 210095, China; pyt_624@163.com (Y.P.); 2022205011@stu.njau.edu.cn (G.W.); 2022105014@stu.njau.edu.cn (M.C.); 2023105026@stu.njau.edu.cn (X.L.); shenm2015@njau.edu.cn (M.S.); 2021105018@stu.njau.edu.cn (H.L.)

**Keywords:** granulosa cell, lactate, HIF-1α, BNIP3, Beclin-1

## Abstract

Background/Objectives: The avascular nature of the follicle creates a hypoxic microenvironment, establishing a niche where granulosa cells (GCs) rely on glycolysis to produce energy in the form of lactate (L-lactate). Autophagy, an evolutionarily conserved stress-response process, involves the formation of autophagosomes to encapsulate intracellular components, delivering them to lysosomes for degradation. This process plays a critical role in maintaining optimal follicular development. However, whether hypoxia regulates autophagy in GCs via lactate remains unclear. Methods: In this study, we investigated lactate-induced autophagy under hypoxia by utilizing glycolysis inhibitors or silencing related genes. Results: We observed a significant increase in autophagy in ovarian GCs under hypoxic conditions, indicated by elevated LC3II levels and reduced P62 levels. Suppressing lactate production through glycolytic inhibitors (2-DG and oxamate) or silencing lactate dehydrogenase (*LDHA/LDHB*) effectively reduced hypoxia-induced autophagy. Further investigation revealed that the HIF1-α/BNIP3/Beclin-1 axis is essential for lactate-induced autophagy under hypoxic conditions. Inhibiting *HIF-1α* activity using siRNAs or PX-478 downregulated BNIP3 expression and subsequently suppressed autophagy. Similarly, *BNIP3* silencing with siRNAs repressed lactate-induced autophagy in hypoxic conditions. Mechanistically, immunoprecipitation experiments showed that BNIP3 disrupted pre-existing Bcl-2/Beclin-1 complexes by competing with Bcl-2 to form Bcl-2/BNIP3 complexes. This interaction released Beclin-1, which subsequently triggered lactate-induced autophagy under hypoxic conditions. Conclusions: These findings unveil a novel mechanism by which hypoxia regulates GC autophagy through lactate production, highlighting its potential role in sustaining follicular development under hypoxic conditions.

## 1. Introduction

In the mammalian ovary, follicular capillaries are restricted to the theca layers located outside the basement membrane, leaving granulosa cells (GCs) and oocytes within an avascular environment. As follicular development advances, the increasing distance between GCs and capillaries further limits oxygen delivery from the blood to the GCs, thereby establishing a hypoxic microenvironment within the follicle [1,2,3]. To adapt to hypoxic stress, cells activate various mechanisms regulated by hypoxia-inducible factor-1α (HIF-1α). This transcription factor induces the expression of genes involved in multiple biological processes, including angiogenesis, energy metabolism, and cell proliferation, enabling cellular adaptation and survival under low-oxygen conditions.

Ovarian follicle development is a highly energy-demanding process. In the hypoxic microenvironment of the follicle, granulosa cells (GCs) rely predominantly on glucose metabolism to satisfy their energy needs [4,5]. This reliance is characterized by increased glycolytic activity, leading to lactate accumulation and the upregulation of key glycolysis-related genes, such as glucose transporters (*GLUT1, GLUT3*), hexokinase (*HK*), pyruvate kinase M2 (*PKM2*), and lactate dehydrogenase (*LDH*) [6,7,8]. Under hypoxic conditions, this metabolic adaptation is further intensified, with increased lactate production in follicles and an accelerated glycolytic rate.

Autophagy, a fundamental, dynamic, and tightly regulated self-degradation process in eukaryotic cells, plays a crucial role in various physiological and developmental processes [9,10]. An increasing number of studies have highlighted the importance of autophagy in follicular development [11,12,13]. Previous research has shown that granulosa cells subjected to hypoxic conditions undergo HIF-1α/BNIP3-mediated autophagy, enabling them to withstand hypoxic stress and support follicular development [14,15]. Additionally, anti-Müllerian hormone (AMH), produced by ovarian granulosa cells, has been implicated in the protection of primordial follicles (PMFs) by inducing ovarian autophagy through the inhibition of FOXO3/FOXO3A phosphorylation [3]. Studies by Tao et al. further demonstrated that autophagy can influence follicular development, including the regulation of granulosa cell lifespan [16]. Nevertheless, whether hypoxia-mediated lactate production contributes to the regulation of autophagy in granulosa cells remains unclear.

Here, we demonstrated that lactate plays a pivotal role in hypoxia-induced autophagy in granulosa cells (GCs). Specifically, we found that hypoxia enhances lactate production, which facilitates the accumulation of HIF-1α, subsequently promoting BNIP3 expression. BNIP3, in turn, initiates autophagy by disrupting the Beclin-1/Bcl-2 complex. Increased binding of BNIP3 to Bcl2 releases Beclin-1 from the original complex, and free Beclin-1 triggers the autophagic process. These findings provide new insights into the regulatory role of lactate in follicular development under hypoxic conditions.

## 2. Materials and Methods

### 2.1. Chemical Reagents

Deoxy-D-glucose (2-DG, S4071), sodium oxamate (oxamate, S6871), α-cyano-4-hydroxycinnamic acid (α-CHCA, S8612), chloroquine (CQ, S6999), and PX-478 (S7612) were purchased from Selleck. Sodium L-lactate (71718) was obtained from Sigma–Aldrich (St. Louis, MO, USA).

### 2.2. Antibodies

Antibodies against TUBA1A (11224-1-AP), Beclin-1 (11306-1-AP), BNIP3 (68091-1-lg), LC3 (14600-1-AP), and LDHA (21799-1-AP) were obtained from Proteintech. Antibodies against SQSTM1/p62 (23214S), BCL-2 (15071T), and HIF-1α (36169S) were purchased from Cell Signaling Technology (Danvers, MA, USA). The antibody against LDHB (PAB698Hu01) was sourced from Cloud–Clone Corp (Houston, TX, USA).

### 2.3. Cell Culture and Treatment

Human ovarian granulosa-like tumor cells (KGN, purchased from YILI Biology, Shanghai, China) were cultured in Dulbecco’s Modified Eagle Medium/Nutrient Mixture F-12 (DMEM-F12) supplemented with 15% fetal bovine serum (FBS) and 1% penicillin–streptomycin. Cells were cultivated under normoxic conditions (37 °C, 5% CO_2_, 21% O_2_) or hypoxic conditions (37 °C, 1% O_2_, 5% CO_2_, 94% N_2_). For drug treatments, cells were exposed to 15 mM 2-deoxyglucose (2-DG), 15 mM oxamate, 50 μM chloroquine (CQ), 2 μM PX-478, 15 mM sodium lactate (Nala), or 3 mM α-cyano-4-hydroxycinnamic acid (CHC) under normoxic or hypoxic conditions for 12 h. In some groups, 15 mM Nala was added 2 h after initial application of the inhibitory drugs, and the cells were incubated for an additional 12 h. For RNA interference experiments, cells were transfected with siRNAs targeting *LDHA, LDHB, HIF-1α, BNIP3*, or scrambled control siRNA for 12 h, followed by incubation with or without 15 mM Nala under normoxic or hypoxic conditions for another 12 h. Autophagy was assessed by comparing the expression of autophagy markers (e.g., LC3-II, p62) between control and treated groups to evaluate the regulatory effects of the treatments.

### 2.4. Small Interfering RNA (siRNA) Transfection

siRNAs targeting *HIF-1α, BNIP3, LDHA, LDHB,* and scrambled control siRNAs (sequences provided in Appendix A) were purchased from GenePharma. Transfection was carried out using Lipofectamine 3000 (Invitrogen, Carlsbad, CA, USA) following the manufacturer’s protocol.

### 2.5. Western Blotting

Collected cells were lysed using cell lysis buffer (Beyotime, Shanghai, China) containing the protease inhibitor PMSF (Beyotime, Shanghai, China). After complete lysis, the lysates were centrifuged at 12,000× *g* for 15 min at 4 °C, and the supernatant was collected. Protein concentrations were determined using the BCA assay kit (Beyotime, Shanghai, China). Protein extracts were separated by sodium dodecyl sulfate-polyacrylamide gel electrophoresis (SDS-PAGE) and transferred to polyvinylidene difluoride (PVDF) membranes (Millipore, Burlington, MA, USA). The membranes were blocked with Tris-buffered saline containing Tween 20 (TBST) and 5% bovine serum albumin (BSA) for 2 h at room temperature to prevent nonspecific binding. The membranes were incubated overnight at 4 °C with primary antibodies diluted in the appropriate buffer. After primary antibody incubation, the membranes were washed and incubated with horseradish peroxidase-conjugated secondary antibodies for 2 h at room temperature. After washing three times with TBST, the developer solution was prepared using WesternBright ECL HRP substrate kit (Advansta, San Jose, CA, USA), and the protein bands were immersed in the developer solution for 30 s. The protein bands were visualized using a fully automated chemiluminescent apparatus, Tanon-5200.

### 2.6. Lactate Concentration Measurement

Lactate levels in granulosa cells (GCs) were measured using a lactate assay kit (A019-2-1, Nanjing Jiancheng Bioengineering Institute, Nanjing, China). Blank, standard, and assay tubes were set up, and distilled water, standard solution, 0.02 mL of the sample to be measured, 1 mL of the prepared enzyme working solution, and 0.2 mL of the color developer were added in order to mix the tubes, and the reaction was terminated by adding 2 mL of termination solution to each tube. The concentration of the chromogenic product was determined by measuring the absorbance value at a wavelength 530 nm. The lactic acid content was calculated using corrected protein concentration results.

### 2.7. Identification of Autophagy

First, cells were transfected using GFP-MAP1LC3B plasmid and cultured for 24 h. Different drug treatments were performed according to the requirements of the experimental subgroups. The Cell Culture and Treatment panel of the Materials and Methods can be referenced for specific drug treatment concentrations. The GFP-MAP1LC3B fusion protein translocates to the autophagosome membrane during autophagy, forming multiple bright green fluorescent puncta visible under fluorescence microscopy. Each punctum corresponds to an autophagosome, allowing quantification of autophagic activity. Cell images were captured using a confocal microscope Zeiss LSM900 (Carl Zeiss AG, Oberkochen, Germany), and the number of green GFP-MAP1LC3B puncta per cell was counted and analyzed using ZEN 3.4 and ImageJ 1.53 to assess autophagic levels.

### 2.8. Co-Immunoprecipitation

Cells were lysed on ice using IP lysis buffer (Beyotime, Shanghai, China) supplemented with PMSF (Beyotime, Shanghai, China). Whole-cell extracts were incubated with specific antibodies for immunoprecipitation at 4 °C for 12 h. Following antibody incubation, 25 μL of Protein A/G Magnetic Beads were added to the mixture, and incubation was continued for 1 h at room temperature. After extensive washing to remove non-specific binding, the immunoprecipitates were analyzed by Western blotting using the appropriate antibodies.

### 2.9. Statistical Analysis

Statistical analysis was conducted using GraphPad Prism 9 software. Each experiment was repeated at least three times with at least three samples to ensure the reliability of the results. Data were presented as the mean ± standard deviation (SD). Group differences were evaluated using one-way analysis of variance (ANOVA), followed by post hoc testing with the least significant difference (LSD) method. A *p*-value less than 0.05 was considered statistically significant.

## 3. Results

### 3.1. Hypoxia Promotes Autophagy in Ovarian Granulosa Cells (GCs) by Stimulating Lactate Production

During follicular development, a hypoxic microenvironment forms within the follicle, encompassing both oocytes and GCs. To determine whether hypoxic conditions promote autophagy in GCs, we first assessed the impact of hypoxia and normoxia on autophagic activity. We evaluated the levels of the autophagy substrate SQSTM1 (p62) and the autophagy marker MAP1LC3B-II (LC3-II). Immunoblot analysis showed that hypoxia led to increased degradation of p62 and accumulation of LC3-II, indicating enhanced autophagy (Appendix A). Additionally, GFP-MAP1LC3B-II (GFP-LC3) transfection followed by hypoxia treatment revealed a significant increase in LC3 punctum formation, consistent with the observed protein expression changes (Appendix A).

Under hypoxic conditions, GCs exhibited active glycolysis, resulting in substantial lactate production (Appendix A). To investigate the role of lactate in hypoxia-induced autophagy, we employed two strategies: (1) treatment with glycolytic inhibitors, 2-deoxy-D-glucose (2-DG) and sodium oxamate (oxamate), and (2) suppression of intracellular lactate production via small interfering RNAs (siRNAs) targeting lactate dehydrogenase isoforms (*LDHA* and *LDHB*).

Immunoblotting revealed that 2-DG and oxamate treatments inhibited the conversion of LC3-I to LC3-II and led to the accumulation of p62 under hypoxia (Figure 1A–C). Fluorescence imaging of GFP-LC3 puncta confirmed a marked reduction in punctum formation after inhibitor treatments (Figure 1D,E). Given the possibility that 2-DG and oxamate might exert effects beyond lactate inhibition, we further silenced *LDHA* and *LDHB* in GCs (Appendix A–J). Simultaneous knockdown of *LDHA* and *LDHB* significantly reduced lactate production, decreased LC3-I to LC3-II conversion, and increased p62 accumulation (Appendix A). Collectively, these findings suggest that hypoxia promotes autophagy in GCs by driving lactate production.

To further confirm lactate’s role in autophagy under hypoxic conditions, we inhibited lactate production and then supplemented lactate exogenously. The addition of lactate effectively restored autophagy, as evidenced by enhanced p62 degradation and LC3-I to LC3-II conversion (Figure 1F–H). Consistently, fluorescence imaging demonstrated an increase in LC3 puncta following lactate supplementation, even after inhibitor treatment (Figure 1I,J). To exclude the effects of non-specific inhibition, we treated GCs with oxamate to inhibit lactate production and chloroquine (CQ) to block autophagic lysosomal fusion and degradation. Subsequent lactate supplementation again restored autophagic activity, corroborating the role of lactate in promoting autophagy (Figure 1K–M). These findings collectively demonstrate that lactate effectively rescues autophagy inhibition and confirm that hypoxia promotes autophagy in GCs by facilitating lactate production.

### 3.2. Lactate Directly Induces GC Autophagy

The results above demonstrate that hypoxia promotes GC autophagy and that lactate positively influences autophagy within the hypoxic microenvironment. To further investigate whether lactate directly affects GC autophagy, we evaluated the impact of exogenous lactate under normoxic conditions. As shown in Figure 2A–E, treatment with exogenous lactate significantly enhanced GC autophagy, as evidenced by decreased p62 expression, increased LC3-II levels, and an elevated number of GFP-LC3 puncta.

To confirm that lactate directly contributes to autophagy, we treated GCs with sodium lactate (Nala) and the monocarboxylate transporter 1 (MCT1) inhibitor α-CHCA, which inhibits lactate uptake and reduces intracellular lactate levels. Sodium lactate treatment notably increased autophagic activity, while α-CHCA effectively suppressed sodium lactate-induced autophagy, as indicated by reduced LC3-II expression, increased p62 accumulation, and fewer GFP-LC3 puncta (Figure 2F–J).

To further validate this observation, chloroquine (CQ) was used to block autophagic lysosomal degradation, allowing us to assess autophagic flux. The results demonstrated that inhibition of lactate uptake by α-CHCA significantly impaired the autophagic pathway, whereas sodium lactate supplementation restored autophagic activity (Figure 2K–M). These findings indicate that sodium lactate directly promotes GC autophagy. Collectively, we conclude that lactate plays an active and crucial role in regulating the autophagic process in GCs.

### 3.3. Lactate Activates the HIF-1α/BNIP3/Beclin-1 Pathway

Previous studies have demonstrated that the HIF-1α/BNIP3/Beclin-1 pathway is a critical signaling axis regulating autophagy in hypoxic cells. However, whether lactate serves as a signaling molecule to activate this pathway remains unclear. In this study, we found that blocking lactate production under hypoxic conditions using 2-deoxy-D-glucose (2-DG) and oxamate significantly reduced the expression of HIF-1α and BNIP3 (Figure 3A,B). To rule out the possibility that these effects were unrelated to lactate inhibition, we further silenced *LDHA* and *LDHB* to suppress lactate production in granulosa cells (GCs). Consistent with the inhibitor results, simultaneous knockdown of *LDHA* and *LDHB* under hypoxia markedly decreased the protein levels of HIF-1α and BNIP3 (Figure 3C,D). These findings suggest that hypoxia activates the HIF-1α/BNIP3/Beclin-1 pathway through lactate.

To further confirm whether hypoxia-induced autophagy is dependent on lactate, we inhibited endogenous lactate production by knocking down *LDHA* and *LDHB*, followed by supplementation with sodium lactate. Lactate supplementation restored the protein expression levels of HIF-1α and BNIP3 under hypoxic conditions (Figure 3E–H). Similarly, under normoxic conditions, exogenous lactate addition also restored HIF-1α and BNIP3 expression, which had been suppressed by lactate inhibition (Figure 3I,J).

Collectively, these results indicate that hypoxia activates the HIF-1α/BNIP3/Beclin-1 pathway in GCs in a lactate-dependent manner.

### 3.4. HIF-1α Facilitates Lactate-Mediated Autophagy in GCs Cultured Under Hypoxia

To further evaluate the role of HIF-1α in regulating autophagy, we inhibited HIF-1α activity through siRNA knockdown or treatment with the inhibitor PX-478. The knockdown efficiency of *HIF-1α* was first confirmed (Appendix A). The results showed that both *HIF-1α* siRNA and PX-478 significantly suppressed the expression of BNIP3 in hypoxic GCs (Figure 4A,C; Appendix A). Additionally, we observed that the reduction in autophagy levels corresponded to the decreased expression of HIF-1α in hypoxic GCs, as indicated by immunoblot analysis of LC3-II and p62 protein levels (Figure 4A–C).

To exclude the possibility of hypoxia-independent lactate production affecting GC autophagy, we directly treated GCs with sodium lactate. Sodium lactate treatment increased BNIP3 expression, whereas inhibition of HIF-1α expression or activity suppressed the lactate-induced upregulation of these proteins. Similarly, inhibition of HIF-1α prevented lactate-induced autophagic flux, as evidenced by reduced LC3-II levels and increased p62 accumulation (Figure 4D–H; Appendix A). These findings collectively suggest that HIF-1α facilitates hypoxia- and lactate-induced autophagy in GCs via the BNIP3/Beclin-1 signaling pathway.

### 3.5. Knockdown of BNIP3 Inhibits Lactate-Induced Autophagy Under Hypoxia

To investigate the role of BNIP3 in lactate-mediated autophagy, we silenced *BNIP3* expression using specific siRNA targeting its transcript (Appendix A). In hypoxic GCs, knockdown of *BNIP3* inhibited autophagy, as evidenced by a reduction in the conversion of LC3-I to LC3-II, increased accumulation of p62 (Figure 5A–C). Similarly, *BNIP3* knockdown suppressed autophagy directly induced by lactate treatment, further supporting the essential role of BNIP3 in lactate-mediated autophagy (Figure 5D–H). Together, these results indicated that the HIF-1α/BNIP3/Beclin-1 pathway is required to mediate the pro-autophagic effects of lactate in GCs under hypoxic conditions.

### 3.6. Hypoxia and Lactate Induce BNIP3 to Disrupt the Bcl-2/Beclin-1 Complex and Activate Beclin-1-Dependent Autophagy

Activation of the HIF-1α/BNIP3 pathway during hypoxia-induced autophagy was evident, but its regulatory mechanism required further investigation. Given the crucial role of Bcl-2 in autophagy regulation, we examined the interactions of Bcl-2 with Beclin-1 and BNIP3 using co-immunoprecipitation (Co-IP) under hypoxic conditions. The results showed that the level of the Bcl-2/BNIP3 complex was significantly higher in hypoxic GCs than in normoxic GCs, suggesting that Beclin-1 was released from the Bcl-2/Beclin-1 complex to induce autophagy (Figure 6A–C). Inhibition of lactate production using 2-DG and oxamate reduced BNIP3 binding to Bcl-2 and increased BNIP3 binding to Beclin-1. This led to a decrease in free Beclin-1 levels and suppression of autophagy (Figure 6D–F).

To further confirm that lactate promotes the release of Beclin-1 from the Bcl-2/Beclin-1 complex via formation of the Bcl-2/BNIP3 complex under hypoxia, we treated GCs directly with lactate. Lactate-treated GCs showed significantly increased formation of the Bcl-2/BNIP3 complex compared to untreated GCs, while the binding of BNIP3 to Beclin-1 showed the opposite trend. This indicates that lactate promotes the release of Beclin-1, which subsequently induces autophagy. Conversely, inhibiting lactate uptake with α-CHCA reduced the formation of the Bcl-2/BNIP3 complex in lactate-treated GCs and increased BNIP3 binding to Beclin-1, suppressing lactate-induced autophagy (Figure 6G–I. These findings suggest that lactate activates HIF-1α to upregulate BNIP3 expression, which disrupts the pre-existing Bcl-2/Beclin-1 complex by competing with Beclin-1 for Bcl-2 binding. The release of Beclin-1 subsequently promoted autophagy in GCs under hypoxic conditions.

## 4. Discussion

Autophagy, a highly conserved cytolytic metabolic mechanism, has been increasingly recognized as essential for oocyte development [4,17], follicular growth and differentiation [18], follicular atresia [11,19], and the reproductive cycle [20]. In vivo experiments have demonstrated that gonadotropins downregulate autophagy in granulosa cells (GCs) [21], and the deletion of key autophagy-related genes, such as *ATG7*, results in ovaries with fewer germ cells and primordial follicles [22,23]. In human ovaries, gonadotropins can suppress autophagy in GCs via activation of the MTOR pathway, which facilitates the orchestration of apoptosis [11,24]. Furthermore, a study by Li et al. revealed a close association between the development of polycystic ovary syndrome (PCOS) and autophagy in ovarian GCs [25]. Given the pivotal role of autophagy in the growth and development of ovarian GCs, understanding its functions is crucial for elucidating the mechanisms regulating GC development and follicular dynamics.

In mammals, cell growth and development depend on oxygen and nutrient transport through the capillary network [26], However, due to the unique physiological structure of the ovary, granulosa cells (GCs) in developing follicles and their differentiated corpus luteum are surrounded by an avascular environment, thus growing under hypoxic conditions [2,27]. Hypoxia regulates catabolism and vascular remodeling around GCs by inducing the expression of hypoxia-inducible factor HIF-1α, activating associated signaling pathways and achieving an adaptive balance to oxygen stress [28]. Simultaneously, hypoxia promotes glycolysis to maintain intracellular metabolic homeostasis [4,29,30]. Increased glycolytic activity leads to lactate accumulation in GCs, yet the relationship between lactate and autophagy in hypoxic GCs remains insufficiently explored. In this study, we investigated the role of lactate in hypoxia-induced autophagy. By utilizing a lactate inhibitor to block lactate production, we found that inhibition of lactate production significantly suppressed autophagy in GCs, highlighting a potential link between lactate accumulation and autophagy under hypoxic conditions.

Due to the hypoxic nature of follicular development, numerous studies have explored the mechanisms regulating autophagy during hypoxia-induced follicular growth. For instance, knockdown of key autophagy-related genes (*ATG5* and *BECN1*) disrupts autophagy, leading to accumulation of the transcription factor WT1 in granulosa cells (GCs). This accumulation, caused by insufficient degradation through the autophagy pathway, inhibits GC differentiation [31]. Tang et al. reported that autophagy mediated by the HIF-1α/BNIP3 pathway effectively promotes GC luteinization and provides protection against granulosa–luteal cell apoptosis [32]. Furthermore, HIF-1α has been shown to inhibit mTORC1 and induce autophagy via AMPK activation [33,34,35]. In addition, HIF-1α activates the transcription of BNIP3, a BH3-domain protein that induces autophagy by competing with Beclin-1 for binding to Bcl-2, thereby releasing Beclin-1 to initiate autophagy [36]. Our study confirmed the involvement of the HIF-1α/BNIP3/Beclin-1 signaling pathway in hypoxia-induced GC autophagy. We observed that deletion of either HIF-1α or BNIP3 significantly inhibited autophagy in GCs. Furthermore, we verified that HIF-1α-mediated upregulation of BNIP3 disrupts the Bcl-2/Beclin-1 complex by competitively binding to Bcl-2. This disruption frees Beclin-1, which subsequently induces autophagy. Notably, our findings revealed that activation of this signaling pathway is entirely dependent on lactate, underscoring the critical role of lactate in GC autophagy under hypoxic conditions. Additionally, it is worth mentioning that a recent study found that hypoxia triggers FoxO1 overactivation via the PI3K/Akt/FoxO pathway, leading to excessive autophagy in GCs and subsequent cell death. This effect was mitigated by melatonin supplementation [37]. Considering the potential for excessive autophagy under hypoxic conditions, future studies could explore the role of lactate in protecting cells from hypoxia-induced excessive autophagy.

Histone lactylation has become a prominent research topic in recent years, particularly since Zhang et al. identified lactate-driven histone lactylation as an epigenetic modification that directly stimulates gene transcription in chromatin [38]. In skeletal muscle studies, Sun et al. proposed that lactate-induced lactylation serves as a critical link between glycolysis and autophagy [39]. Similarly, in research on intervertebral disc degeneration, Zhang et al. reported that inhibiting AMPKα lactylation effectively promotes autophagy and delays the senescence of related cells [40]. Furthermore, Huang et al. demonstrated that lactylation stabilizes transcription factor EB (TFEB), a key regulator of autophagy and lysosomal gene expression, thereby promoting elevated autophagy levels [41]. These findings suggest that histone lactylation likely plays a significant role in cellular autophagy regulation, potentially serving as an essential mechanism linking metabolic changes to autophagic processes.

Our study demonstrates the regulation of autophagy in hypoxic granulosa cells (GCs) by lactate through in vitro experiments. However, the role of lactylation in autophagy under hypoxic conditions remains unclear. Given that lactate serves as a substrate for histone lactylation, future research could focus on the role of histone lactylation in the autophagic process of GCs. Specifically, we aim to identify the gene loci where lactylation modifications occur, elucidate the functional significance of lactylation in GC autophagy, and explore the broader implications of histone modifications in cellular autophagy. Similarly, our study only preliminarily found that lactate promotes autophagy through the HIF-1α/BNIP3/Beclin-1 signaling axis under hypoxia, and further exploration can be carried out regarding the activation and transmission of this signaling axis. In addition, cancer cells are inherently part of a hypoxic process during development as they preferentially use glycolytic metabolism to maintain their own development. It would have been more precise to use primary granulosa cells for hypoxia treatment in this study, which would have compelled us to later explore how hypoxia promotes GCs autophagy. In order to strengthen the persuasiveness of our findings, we plan to establish relevant animal models and conduct in vivo experiments in the future. These approaches will provide deeper insights into the mechanisms underlying lactylation and its impact on autophagy, bridging the gap between in vitro observations and physiological relevance.

Overall, our current study robustly demonstrates that hypoxia activates lactate-dependent autophagy in granulosa cells (GCs), elucidates the underlying mechanism by which lactate regulates GC autophagy under hypoxic conditions, and provides novel insights into the role of autophagy in follicular development. Furthermore, these findings offer potential therapeutic strategies for future research on ovary-related diseases.

## 5. Conclusions

This study demonstrates that lactate effectively promotes granulosa cell autophagy via the HIF-1α/BNIP3/Beclin-1 signaling pathway. By employing lactate inhibition and supplementation strategies, we have elucidated the regulatory mechanism of autophagy in granulosa cells from the perspective of lactate metabolism (Figure 7). These findings not only enhance our understanding of autophagy regulation in granulosa cells but also provide a novel avenue for exploring potential therapeutic approaches for ovarian diseases in women.

## Figures and Tables

**Figure 1 genes-16-00014-f001:**
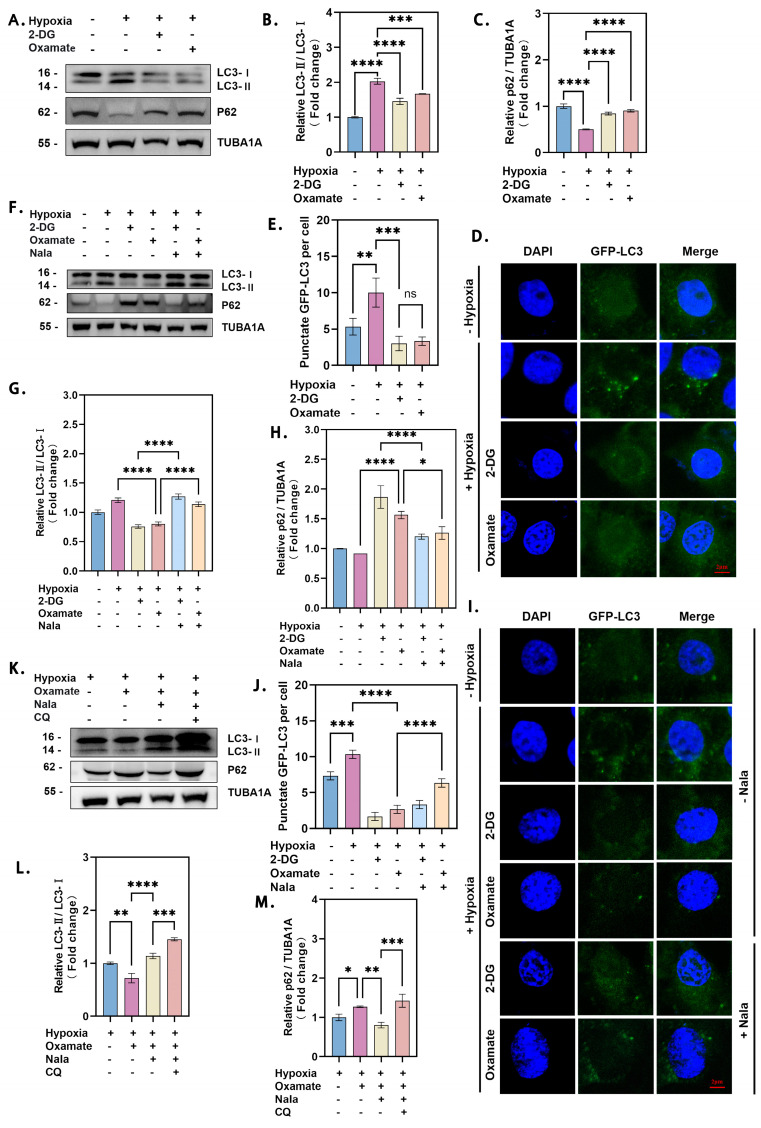
Hypoxia promotes autophagy of ovarian GCs through stimulation of lactate production. (**A**) GCs were treated with 15 mM 2-DG or 15 mM oxamate for 2 h, followed by 12 h of hypoxia, and protein levels of LC3 and p62 were determined by Western blot. (**B**) Quantitative analysis showed a significant increase in LC3-I to LC3-II conversion and (**C**) a decrease in p62 levels, with data presented as the mean ± SD (*n* ≥ 3, **** *p* < 0.0001; *** *p* < 0.001). (**D**) GFP-LC3 plasmid transfection revealed increased puncta under hypoxia, visualized by confocal microscopy, and (**E**) the number of GFP-LC3 puncta per cell was significantly elevated, with data from at least 5 cells per group (*** *p* < 0.001; ** *p* < 0.01; ns, not significant). (**F**) Cells treated with 15 mM 2-DG or oxamate, with or without 15 mM Nala supplementation after 2 h, and cultured under hypoxia or normoxia for 12 h showed altered LC3 and p62 levels by Western blot. (**G**) Quantification indicated a significant effect of Nala on LC3-I to LC3-II conversion and (**H**) p62 reduction (*n* ≥ 3, **** *p* < 0.0001; * *p* < 0.05). (**I**) Immunofluorescence confirmed GFP-LC3 punctum localization, and (**J**) punctum quantification showed consistent results. At least 5 cells were counted per group (**** *p* < 0.0001; *** *p* < 0.001). (**K**) Cells treated with 15 mM oxamate and 50 μM CQ, with or without 15 mM Nala after 2 h, under hypoxia for 12 h displayed significant changes in LC3 and p62 levels by Western blot, with quantitative analysis showing effects on (**L**) LC3-I to LC3-II conversion and (**M**) p62 levels (n ≥ 3,**** *p* < 0.0001; *** *p* < 0.001; ** *p* < 0.01; * *p* < 0.05). Error bars represent the standard deviation of the mean.

**Figure 2 genes-16-00014-f002:**
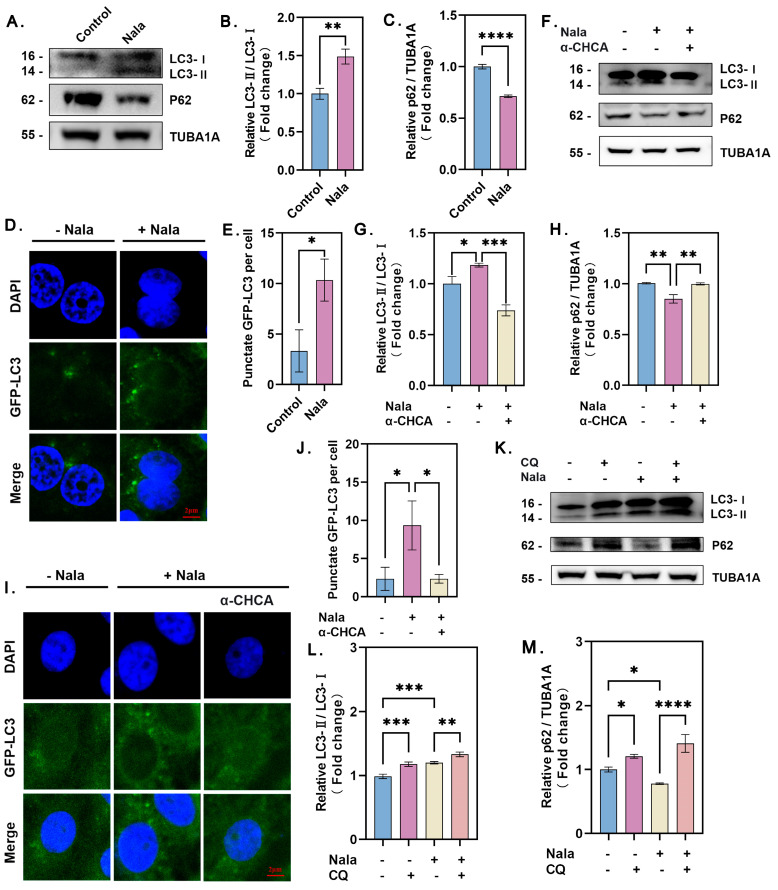
Lactate actively induces autophagy in GCs. (**A**) Cells were cultured under normoxia for 12 h with or without the addition of 15 mM Nala. Intracellular LC3 and p62 protein levels were detected by Western blot. (**B**) Quantitative analysis of LC3-I to LC3-II conversion and (**C**) decreases in p62 levels, with data presented as the mean ± SD; *n* ≥ 3; **** *p* < 0.0001; ** *p* < 0.01. (**D**) Immunofluorescence localization of GFP-LC3 in granulosa cells. (**E**) The number of GFP-LC3 puncta per cell was quantified. At least 5 cells were counted per group, and data are presented as the mean ± SD; * *p* < 0.05. (**F**) Protein levels of LC3 and p62 were assessed after inhibition of lactate with 3 mM α-CHCA, with or without the addition of 15 mM Nala, and quantified. (**G**) Quantitative analysis of LC3-I to LC3-II conversion. Data are presented as the mean ± SD; *n* ≥ 3; *** *p* < 0.001; * *p* < 0.05. (**H**) Quantitative analysis of p62 protein levels. Data are presented as the mean ± SD; ** *p* < 0.01. (**I**) Immunofluorescence localization of GFP-LC3 in granulosa cells. (**J**) The number of GFP-LC3 puncta per cell was quantified. Data are presented as the mean ± SD; * *p* < 0.05. At least 5 cells were counted per group. (**K**) After 50 μM CQ treatment for 2 h, cells were cultured under normoxia with or without 15 mM Nala. Protein levels of LC3 and p62 were detected and quantified. (**L**) Quantitative analysis of LC3-I to LC3-II conversion. Data are presented as the mean ± SD; *n* ≥ 3; *** *p* < 0.001; ** *p* < 0.01. (**M**) Quantitative analysis of p62 protein levels. Data are presented as the mean ± SD; *n* ≥ 3; **** *p* < 0.0001; * *p* < 0.05. Error bars represent the standard deviation of the mean.

**Figure 3 genes-16-00014-f003:**
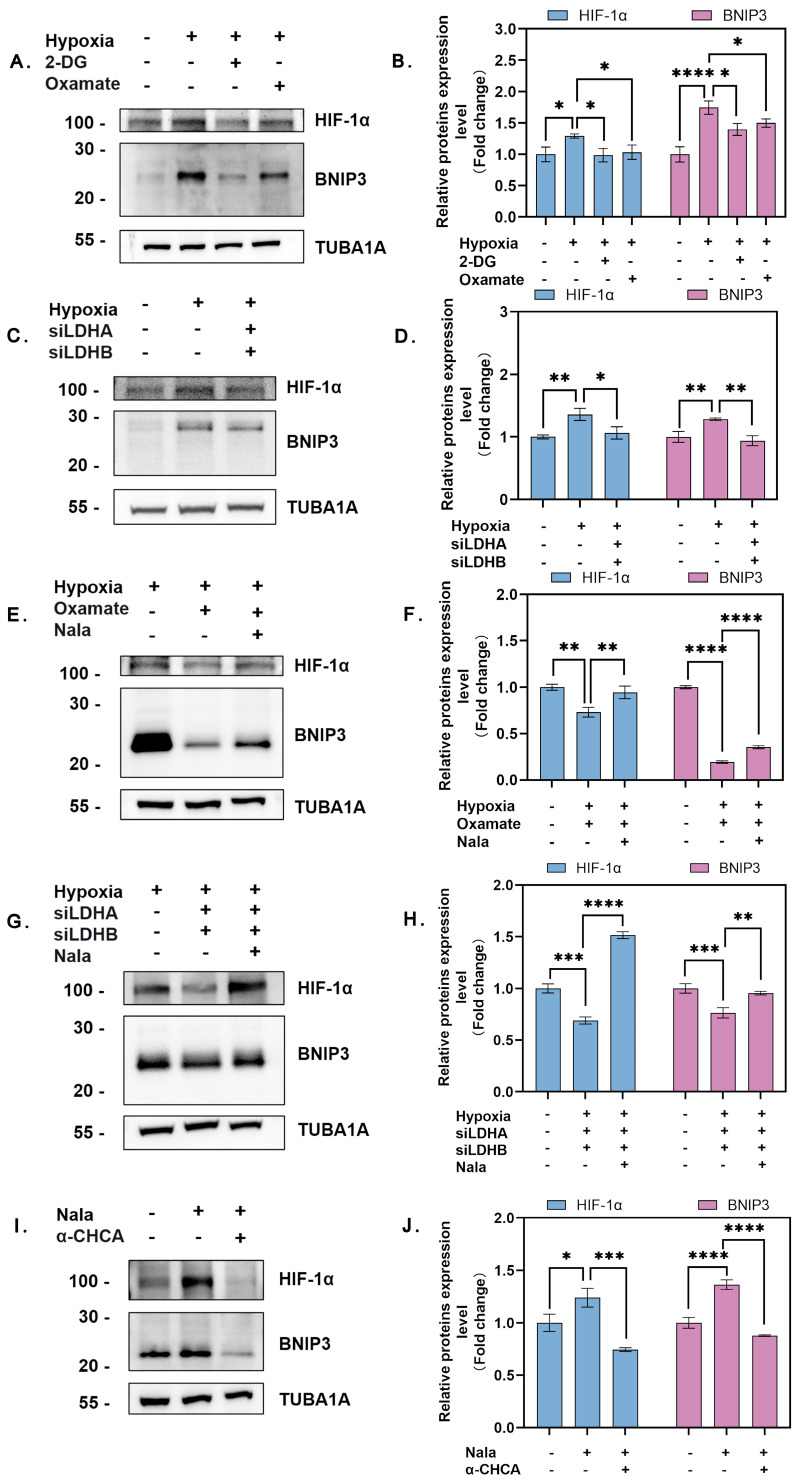
Lactate promotes activation of the HIF-1α/BNIP3/Beclin-1 pathway. (**A**) GCs were treated with 15 mM 2-DG or 15 mM oxamate and exposed to hypoxia or normoxia for 12 h. Protein levels of HIF-1α and BNIP3 were detected by Western blot and quantified. (**B**) Quantitative analysis of HIF-1α and BNIP3 protein levels. Data represent the mean ± SD; *n* ≥ 3; **** *p* < 0.0001; * *p* < 0.05. (**C**) Cells were transfected with or without siRNAs targeting *LDHA* and *LDHB* for 12 h and then exposed to different oxygen conditions. Protein levels of HIF-1α and BNIP3 were detected and quantified. (**D**) Quantitative analysis of HIF-1α and BNIP3 protein levels. Data represent the mean ± SD; *n* ≥ 3; ** *p* < 0.01; * *p* < 0.05. (**E**) GCs were treated with 15 mM 2-DG or 15 mM oxamate for 2 h with or without 15 mM Nala and then cultured under hypoxia or normoxia for 12 h. Protein levels of HIF-1α and BNIP3 were detected and quantified. (**F**) Quantitative analysis of HIF-1α and BNIP3 protein levels. Data represent the mean ± SD; *n* ≥ 3; **** *p* < 0.0001; ** *p* < 0.01. (**G**) After silencing LDHA and LDHB, with or without the addition of 15 mM Nala, cells were cultured under hypoxia. Protein levels of HIF-1α and BNIP3 were detected and quantified. (**H**) Quantitative analysis of HIF-1α and BNIP3 protein levels. Data represent the mean ± SD; *n* ≥ 3; **** *p* < 0.0001; *** *p* < 0.001; ** *p* < 0.01. (**I**) Cells were treated with 3 mM α-CHCA for 2 h to inhibit lactate uptake, followed by the addition of 15 mM Nala, and cultured under hypoxia. Protein levels of HIF-1α and BNIP3 were detected and quantified. (**J**) Densitometric analysis of HIF-1α and BNIP3 protein levels. Data represent the mean ± SD; *n* ≥ 3; **** *p* < 0.0001; *** *p* < 0.001; * *p* < 0.05. Error bars represent the standard deviation of the mean.

**Figure 4 genes-16-00014-f004:**
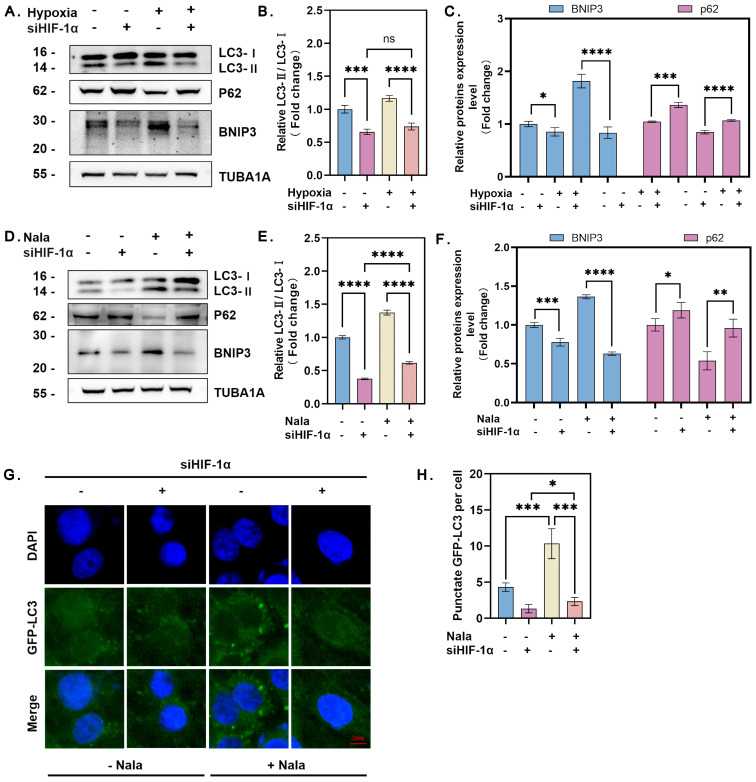
HIF-1α enhances lactate-driven autophagy in GCs under hypoxia. (**A**) GCs were transfected with or without HIF-1α siRNA for 12 h and then cultured under hypoxia or normoxia. Protein levels of LC3, p62, and BNIP3 were detected by Western blot. (**B**) Quantitative analysis of LC3-I to LC3-II conversion. Data represent the mean ± SD; *n* ≥ 3; **** *p* < 0.0001; *** *p* < 0.001; ns, not significant. (**C**) Quantitative analysis of BNIP3 and p62 protein levels. Data represent the mean ± SD; *n* ≥ 3; **** *p* < 0.0001; *** *p* < 0.001; * *p* < 0.05. (**D**) Cells were transfected with or without HIF-1α siRNA for 12 h, treated with or without 15 mM Nala, and cultured for an additional 12 h. Protein levels of LC3, p62, and BNIP3 were analyzed. (**E**) Quantitative analysis of LC3-I to LC3-II conversion. Data represent the mean ± SD; *n* ≥ 3; **** *p* < 0.0001. (**F**) Quantitative analysis of BNIP3 and p62 protein levels. Data represent the mean ± SD; *n* ≥ 3; **** *p* < 0.0001; *** *p* < 0.001; ** *p* < 0.01; * *p* < 0.05. (**G**) Immunofluorescence localization of GFP-LC3 in granulosa cells. (**H**) Quantification of GFP-LC3 puncta per cell. Data represent the mean ± SD; *** *p* < 0.001; * *p* < 0.05. At least 5 cells were counted per group. Error bars represent the standard deviation of the mean.

**Figure 5 genes-16-00014-f005:**
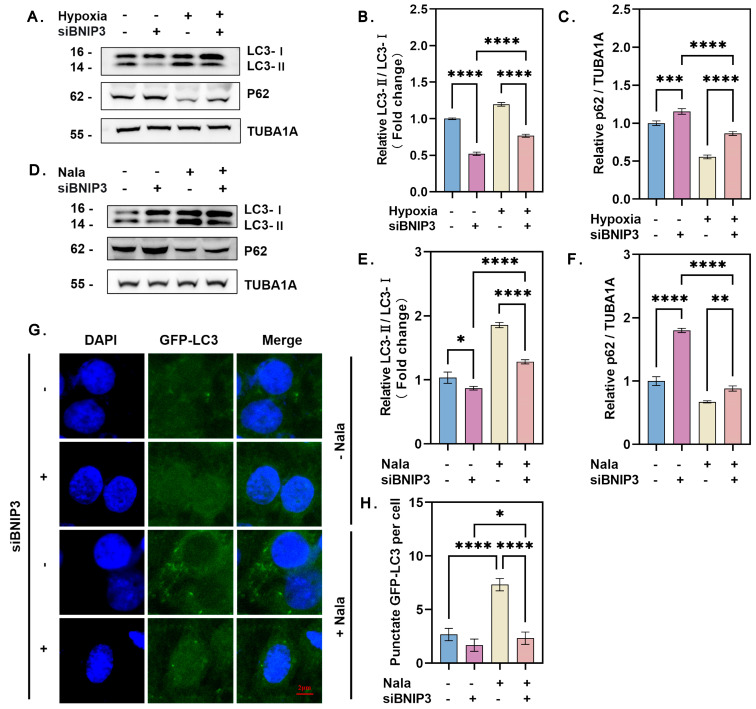
Knockdown of BNIP3 inhibits lactate-induced autophagy under hypoxia. (**A**) GCs were transfected with or without BNIP3 siRNA for 12 h and then cultured under hypoxia or normoxia. Protein levels of LC3 and p62 were detected by Western blot. (**B**) Quantitative analysis of LC3-I to LC3-II conversion. Data represent the mean ± SD; *n* ≥ 3; **** *p* < 0.0001. (**C**) Quantitative analysis of p62 protein levels. Data represent the mean ± SD; *n* ≥ 3; **** *p* < 0.0001; *** *p* < 0.001. (**D**) GCs transfected with or without BNIP3 siRNA for 12 h were treated with or without 15 mM Nala and cultured for an additional 12 h. Protein levels of LC3 and p62 were analyzed. (**E**) Quantitative analysis of LC3-I to LC3-II conversion. Data represent the mean ± SD; *n* ≥ 3; **** *p* < 0.0001; * *p* < 0.05. (**F**) Quantitative analysis of p62 protein levels. Data represent the mean ± SD; *n* ≥ 3; **** *p* < 0.0001; ** *p* < 0.01. (**G**) Immunofluorescence localization of GFP-LC3 in GCs. (**H**) Quantification of GFP-LC3 puncta per cell. Data represent the mean ± SD; **** *p* < 0.0001; * *p* < 0.05. At least 5 cells were counted per group. Error bars represent the standard deviation of the mean.

**Figure 6 genes-16-00014-f006:**
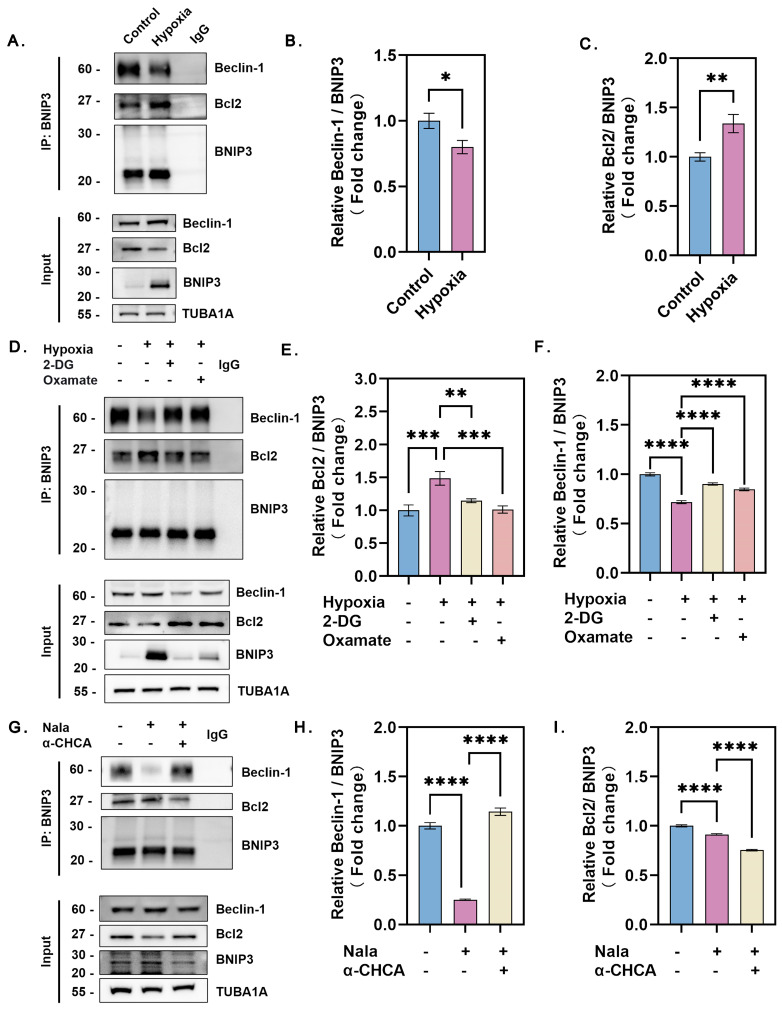
Hypoxia and lactate trigger BNIP3 to disrupt the Bcl-2/Beclin-1 complex, activating Beclin-1-dependent autophagy. (**A**) Cells cultured under hypoxia or normoxia were immunoprecipitated with an anti-BNIP3 antibody, and the precipitates were analyzed by immunoblotting to assess protein expression levels. (**B**) Quantitative analysis of Beclin-1 protein levels. Data represent the mean ± SD; *n* ≥ 3; * *p* < 0.05. (**C**) Quantitative analysis of Bcl-2 protein levels. Data represent the mean ± SD; *n* ≥ 3; ** *p* < 0.01. (**D**) Cells were treated with 15 mM 2-DG or 15 mM oxamate and exposed to hypoxia or normoxia for 2 h. Following treatment, cells were immunoprecipitated using an anti-BNIP3 antibody, and protein expression levels were analyzed by immunoblotting. (**E**) Quantitative analysis of Beclin-1 protein levels. Data represent the mean ± SD; *n* ≥ 3; *** *p* < 0.001; ** *p* < 0.01. (**F**) Quantitative analysis of Bcl-2 protein levels. Data represent the mean ± SD; *n* ≥ 3; **** *p* < 0.0001. (**G**) After lactate inhibition using 3 mM α-CHCA for 2 h, cells were supplemented with Nala and cultured under normoxia for 12 h. Cells were immunoprecipitated with an anti-BNIP3 antibody, and protein expression levels of the precipitates were analyzed by immunoblotting. (**H**) Quantitative analysis of Beclin-1 protein levels. Data represent the mean ± SD; *n* ≥ 3; **** *p* < 0.0001. (**I**) Quantitative analysis of Bcl-2 protein levels. Data represent the mean ± SD; *n* ≥ 3; **** *p* < 0.0001. Error bars represent the standard deviation of the mean.

**Figure 7 genes-16-00014-f007:**
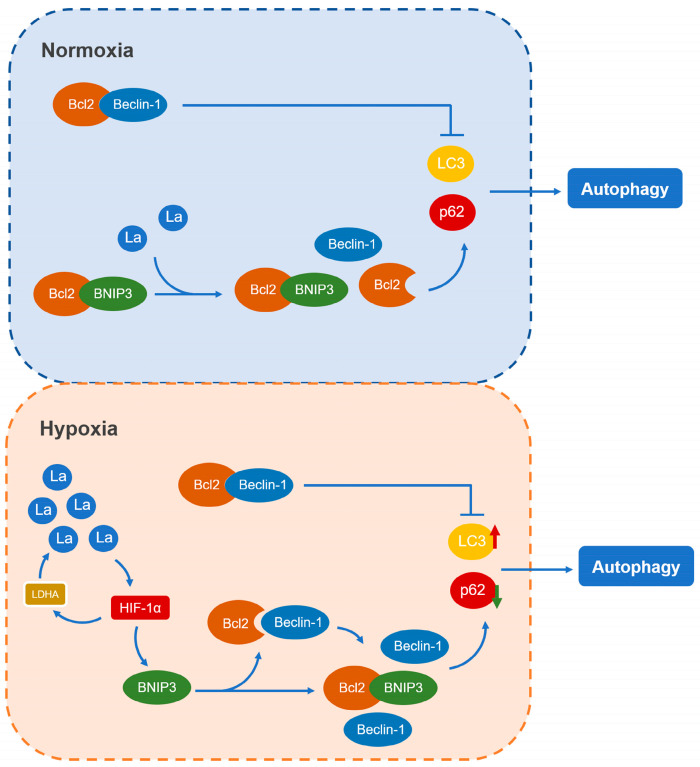
Schematic diagram illustrating the molecular mechanism by which lactate promotes granulosa cell autophagy via the HIF-1α/BNIP3/Beclin-1 signaling pathway. Under hypoxic conditions, lactate activates the HIF-1α/BNIP3/Beclin-1 pathway, leading to upregulation of BNIP3. BNIP3 promotes granulosa cell autophagy by competitively binding to Bcl-2, thereby increasing the availability of free Beclin-1 to initiate the autophagic process.

## Data Availability

The original contributions presented in the study are included in the article/Appendix A, further inquiries can be directed to the corresponding author.

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
