# Peer review of "Lactate Promotes Hypoxic Granulosa Cells’ Autophagy by Activating the HIF-1α/BNIP3/Beclin-1 Signaling Axis"

_genes, 2024, doi:10.3390/genes16010014_

Round 1
Reviewer 1 Report
Comments and Suggestions for Authors
In the present work, Pan et al. convincingly demonstrate the role lactate plays in granulosa cell autophagy. There are a few minor revisions requested:
1. This study uses a cancer cell line as a model of human granulosa cells. The title should be changed to include "human", and preferably "KGN", to make the species and system clear. e.g., "Lactate promotes human granulosa cells (KGN) autophagy by activating the HIF-1α/BNIP3/Beclin-1 signaling axis under hypoxia". The authors discuss the model somewhat (lines 428-38) but more discussion should be included on limitations, as cancer cells may have altered autophagy and hypoxia pathways.
2. The authors should also discuss the oxygenation environment of the follicle and how it relates to "normoxic" conditions, which in this study is 21% (ambient) O2 -- likely far higher than "normal" oxygenation of the GC compartment of the follicle? Similarly, how was 5% chosen as hypoxic?
3. There is no supplementary figures available for me to review. However, I think measured lactate would be interesting data to include in a main figure.
4. The conflicting results in Fig 1 for p62 should be discussed. In Fig 1H, the addition of Nala to hypoxia+oxamate increases p62 levels, while in Fig 1M, this same combination decreases p62 levels.
5. I found the final schematic figure confusing. The slightly dissociated Bcl2/Beclin-1 complex in the centre seems redundant as in this figure Beclin-1 is the driver of autophagy. In the hypoxic condition, the arrow from Beclin-1 is replaced by a solid line, reinforcing this point, but changing the style from the normoxic condition. Finally, it would be nice to include p62 and LC3 in the figure, if even as markers of autophagy (i.e., on the right side of the figure), since they are examined in detail in this research.
Author Response
尊敬的评论者:
请参阅附件。

Reviewer 2 Report
Comments and Suggestions for Authors
The main topic of the paper written by Yitong Pan et al. is the evaluation of the role of lactate in the process of autophagy of GCs in a hypoxic condition.
Previous findings suggest that granulosa cells undergo HIF-1a – BNIP3 – mediated autophagy in hypoxia conditions.Studies concerning the cellular adaptation mechanisms under hypoxic conditions, highlighted the lactate accumulation. Furthermore, other research shows the link between glycolysis and autophagy through the process of histone lactylation in specific conditions. However, how the presence of lactate under hypoxia was related to autophagia was still unclear. Through these considerations and the comparison with the other papers cited in the introduction and in the discussion, is highlighted the lack of knowledge that led them to pursuing these studies and the high contribution to the existing knowledge, which is for sure a strength of this paper.
Although, to make the study more comprehensible by the reader, I think there should be a little introduction/explanation about the autophagy mechanism controlled and mediated by the HIF-1a/BNIP3/Beclin – 1.
For what concern materials and methods, this section is well organized through a division in paragraphs, which make it easier the comprehension. All the materials (including the catalogue numbers of RRIDs for all the antibodies present) and the software used to perform the statistical analysis are well described. However, it is not defined the time frame of the study and is not defined the number of samples (human ovarian granulosa – like tumore cells) considered. Moreover, the section that reports how the identification of autophagy was performed, talks about different treatments that were executed on the cells trasferred with the plasmid, without describing them. These considerations can limit the reproducibility of the study.
The results are structured in different and specific sections that are helpful for a better comprehension.
Each portion is followed by its specific graphics: this way of structure make the interpretation and apprehention of the results.
The limitation associated to a study in vitro are taken in consideration by the authors, who suggest in vivo experiments and use of relevant animal models to increase the validity of their findings. It can be stated the evidence supports the conclusion.
Keywords: The keywords provided in the study, such as “Granulosa cells”, “Hipoxia”, “lactate”, “BNIP3” and “autophagy” are essential to be able to focus on the main topics of the paper. However, by adding also “Beclin – 1” and “HIF-1a” the paper would be more accessible to researchers that are conducting studies in the same field.
Graphics: the graphics are well representative about the results obtaneid. Althought, I think that should be reconsidered their organization.
There is a specific section concerning acknowledge but are not cited the ethical consensus.
Reviewer 3 Report
Comments and Suggestions for Authors
Comments about the manuscript:
“Lactate promotes granulosa cells autophagy by activating the HIF-1α/BNIP3/Beclin-1 signaling axis under hypoxia”
Autophagy is manifested by the formation of autophagosomes that encapsulate intracellular components, which are then degraded by lysosomes. Furthermore, in the ovarian follicle, a hypoxic environment, granulosa cells depend on glycolysis to produce energy. However, the question still remains whether hypoxia regulates autophagy in granulosa cells and the aim of the study presented here was to attempt to answer this question. To do this, the authors cultured granulosa cells from a commercial strain, to which they applied several treatments. The results obtained reveal a new mechanism by which hypoxia regulates autophagy of granulosa cells via the production of lactate, characteristic of a hypoxic environment.
This work, which brings new elements to the knowledge of the regulation of autophagy by hypoxic conditions in the particular case of granulosa of ovarian follicles, seems to me to be very well done. It needs some improvements to the manuscript, however.
General: there are a lot of abbreviations. A list of them might be helpful (but not required).
Use italics to write the names of genes.
Materials and methods
Page 2, line 85. “based on the experimental design.”: what is this experimental design? Please specify here.
Page 3, line 99, line 116. “following the manufacturer’s protocol.” is not sufficient for a scientific article, please briefly describe the method used.
Page 3, line 113. “using the WesternBright ECL HRP”: briefly describe the method.
Page 17, figure 7. I really like this synthetic figure, but it is not mentioned in the text.
Page 3, line 127. “using appropriate graphical software”: please clarify what "appropriate software" means, and what it is?
